# Two Loci Contribute to Age-Related Hearing Loss Resistance in the Japanese Wild-Derived Inbred MSM/Ms Mice

**DOI:** 10.3390/biomedicines10092221

**Published:** 2022-09-07

**Authors:** Shumpei P. Yasuda, Yuki Miyasaka, Xuehan Hou, Yo Obara, Hiroshi Shitara, Yuta Seki, Kunie Matsuoka, Ai Takahashi, Eri Wakai, Hiroshi Hibino, Toyoyuki Takada, Toshihiko Shiroishi, Ryo Kominami, Yoshiaki Kikkawa

**Affiliations:** 1Deafness Project, Department of Basic Medical Sciences, Tokyo Metropolitan Institute of Medical Science, Tokyo 156-8506, Japan; 2Division of Experimental Animals, Graduate School of Medicine, Nagoya University, Nagoya 466-8550, Japan; 3Graduate School of Medical and Dental Sciences, Niigata University, Niigata 951-8510, Japan; 4Graduate School of Life and Environmental Sciences, University of Tsukuba, Tsukuba 305-8572, Japan; 5Laboratory for Transgenic Technology, Center for Basic Technology Research, Tokyo Metropolitan Institute of Medical Science, Tokyo 156-8506, Japan; 6Division of Glocal Pharmacology, Department of Pharmacology, Graduate School of Medicine, Osaka University, Osaka 565-0871, Japan; 7Integrated Bioresource Information Division, RIKEN BioResource Research Center, Tsukuba 305-0074, Japan; 8RIKEN BioResource Research Center, Tsukuba 305-0074, Japan

**Keywords:** age-related hearing loss, C57BL/6J strain, MSM/Ms strain, consomic strain, congenic strain, *ahl* loci

## Abstract

An MSM/Ms strain was established using Japanese wild mice, which exhibit resistance to several phenotypes associated with aging, such as obesity, inflammation, and tumorigenesis, compared to common inbred mouse strains. MSM/Ms strain is resistant to age-related hearing loss, and their auditory abilities are sustained for long durations. The age-related hearing loss 3 (*ahl3*) locus contributes to age-related hearing in MSM/Ms strain. We generated *ahl3* congenic strains by transferring a genomic region on chromosome 17 from MSM/Ms mice into C57BL/6J mice. Although C57BL/6J mice develop age-related hearing loss because of the *ahl* allele of the cadherin 23 gene, the development of middle- to high-frequency hearing loss was significantly delayed in an *ahl3* congenic strain. Moreover, the novel age-related hearing loss 10 (*ahl10*) locus associated with age-related hearing resistance in MSM/Ms strain was mapped to chromosome 12. Although the resistance effects in *ahl10* congenic strain were slightly weaker than those in *ahl3* congenic strain, slow progression of age-related hearing loss was confirmed in *ahl10* congenic strain despite harboring the *ahl* allele of cadherin 23. These results suggest that causative genes and polymorphisms of the *ahl3* and *ahl10* loci are important targets for the prevention and treatment of age-related hearing loss.

## 1. Introduction

Mouse genetics is an important approach for the clarification of the basic genetic and biological mechanisms underlying the development of hearing loss in humans [1,2]. Many genes associated with human hearing loss have been identified using forward genetic approaches from spontaneous and n-ethyl-n-nitrosourea-induced mutants, and reverse genetic approaches have successfully been used to generate animal models that help understand the auditory phenotypes and molecular mechanisms underlying the development of hearing loss.

A foundation for the mouse genetic approach to hearing loss is the inbred strain, which includes the mating of siblings for 20 or more consecutive generations. It is estimated that an average of 98.6% of the loci are homozygous in the inbred strain of the 20th generation [3]. The homozygosity of most loci leads to a decrease and elimination of phenotypic heterogeneity and consequently reduces the dispersion of data and experimental perturbations in both wild-type and mutant mice. The reduction in genetic and phenotypic heterogeneity is particularly advantageous for a forward genetics approach to hearing loss [1,2]. The identification of susceptibility genes associated with mild to moderate hearing loss phenotypes, such as those associated with age-related hearing loss (ARHL) and noise-induced hearing loss (NIHL), is very difficult in humans using forward genetics approaches, potentially because of high genetic heterogeneity. However, the forward genetics approach in mice can reduce the genetic background effects in hearing loss.

Moreover, hearing loss in inbred strains is well-characterized phenotypically and divergent in terms of severity and onset time. For example, the C57BL/6J (B6) strain is a well-known mouse model of late-onset, progressive ARHL that develops severe hearing loss with respect to stimuli at high frequency at 4–6 months of age, middle frequency at 8–12 months of age, and low frequency at 16–20 months of age [4,5,6]. The DBA/2J (DBA2) strain is a model of early-onset and progressive hearing loss, wherein hearing loss progresses from high to low frequencies, with hearing loss at 1 month of age at high frequency (32 kHz), 4 months at middle frequency (16 kHz), and 7 months at low frequency (8 kHz) [5,7,8]. Previous studies have confirmed strain-specific susceptibility alleles and loci associated with the onset and severity of hearing loss in the B6 and DBA2 strains. The *ahl* (*Cdh23^ahl^*: c.753G>A) allele of cadherin 23 gene (*Cdh23*) is a common risk factor for ARHL development in both strains [4,8,9]. The *Cdh23^ahl^* allele was identified as a base before the splice-donor site of exon 9 of the *Cdh23* gene, leading to partial skipping of a single exon, and was shown to be majorly responsible for ARHL in multiple inbred mouse strains [9]. Our previous study confirmed the development of ARHL, age-related degeneration of cochlear stereocilia, and loss of hair cells via substitution of the wild-type (*Cdh23*^+^) allele with the *Cdh23^ahl^* allele using genome editing in B6 mice [6,10]. The DBA2 strain carries other susceptibility alleles, which can accelerate the onset and increase severity of hearing loss. The DBA2 mice exhibit progressive shortening of stereocilia due to additive effects between the *Cdh23^ahl^* allele and the *ahl8* (*Fscn2^ahl8^*: p.Arg109His) allele of fascin actin-bundling protein 2 gene (*Fscn2*) [8,11,12]. A susceptibility allele on chromosome 5 of the DBA2 strain is associated with the severity of mid-frequency-specific ARHL [7,13]. The presence of strain-specific alleles associated with hearing loss has also been reported in other inbred strains [1,2], suggesting that inbred mouse strains are powerful bioresources for identifying novel susceptibility genes for hearing loss.

In contrast, the MSM/Ms (MSM) strain, an inbred strain derived from the Japanese wild mouse, *Mus musculus molossinus* [14,15,16], is resistant to the development of hearing loss. It exhibits resistance to several phenotypes associated with aging, such as obesity [16,17], inflammation [18], and tumorigenesis [19,20], when compared with common inbred mouse strains. MSM strain is resistant to ARHL and possesses long-lasting auditory abilities [6,21]. Moreover, our previous study reported that a C57BL/6J-Chr 17^MSM^/Ms (B6-Chr17^MSM^) consomic strain, in which the chromosome 17 pair of the recipient B6 strain was replaced with the corresponding chromosome 17 pair of the donor MSM strain, maintained hearing ability over long durations [21]. Notably, the development of ARHL in the B6-Chr17^MSM^ strain was suppressed despite the presence of the *Cdh23^ahl^* allele on chromosome 10, suggesting that the substitution of chromosome 17 counteracts the susceptibility effects of the *Cdh23^ahl^* allele. Furthermore, using linkage analysis, we suggested that an age-related hearing loss 3 (*ahl3*) locus on chromosome 17 was potentially associated with ARHL resistance in B6-Chr17^MSM^ strain [21].

The *Cdh23^ahl^* allele has been reported to have extremely strong susceptibility effects on ARHL onset in mice because ARHL has been observed in all inbred strains carrying the *Cdh23^ahl^* allele [9]. Therefore, the identification of genetic factors that play a role in counteracting and reducing the susceptibility effects of the *Cdh23^ahl^* allele in ARHL may contribute to the prevention and therapy of ARHL. In this study, we performed detailed hearing tests by recording the auditory brainstem response (ABR) and distortion product otoacoustic emission (DPOAE) in the B6-Chr17^MSM^ consomic and *ahl3* congenic strains. The results confirmed that the *ahl3* locus is associated with ARHL resistance in middle- to high-frequency sounds. Moreover, we identified that a C57BL/6J-Chr 12C^MSM^/Ms (B6-Chr12C^MSM^) subconsomic strain, in which the homologous centromeric region (69.8 Mb) on chromosome 12 of the B6 strain is replaced by the corresponding region of the MSM strain, is resistant to ARHL and that a novel age-related hearing loss 10 (*ahl10*) locus on chromosome 12 contributes to ARHL resistance in the B6-Chr12C^MSM^ strain.

## 2. Materials and Methods

### 2.1. Mice

The B6 (C57BL/6JJcl) strain was purchased from CLEA Japan (Tokyo, Japan). The MSM, B6-Chr17^MSM^ consomic, and B6-Chr12C^MSM^ subconsomic strains were obtained from the National Institute of Genetics. The B6-Cdh23^+/+^ strain was produced by genome editing as previously reported [10]. All congenic lines were produced by intercrossing mice possessing the genomic region of interest on chromosomes 17 or 12, selected from the (B6 × B6-Chr17^MSM^) F_2_ and (B6 × B6-Chr12C^MSM^) F_2_ mice. Genomic regions were genotyped using simple sequence length polymorphism (SSLP) markers and single nucleotide polymorphism (SNP) markers. PCR templates for genotyping were extracted from the pinnae (diameter: 2 mm) of adult mice anesthetized with isoflurane using the KAPA Express Extract Kit (Kapa Biosystems, Wilmington, MA, USA). PCR amplification was conducted using a KAPA2G Fast PCR Kit (Kapa Biosystems) under the following conditions: one cycle at 95 °C for 2 min, followed by 40 cycles at 95 °C for 15 s, 58 °C for 20 s, and 72 °C for 5 s. The amplicons were subjected to 4% agarose gel electrophoresis and stained with ethidium bromide. The primers used for SSLP genotyping are listed in Appendix A. SNP genotyping was performed using a high-resolution melting (HRM) assay, using the LightCycler 480 High-Resolution Melting Master (Roche Diagnostics, Mannheim, Germany) with the primers listed in Appendix A. The primers were designed to be 25-bp-long and bind to both sides of the target SNPs. The 6 μL mixtures of for the HRM assay contained 1× LightCycler 480 High-Resolution Melting Master Mix, 2.5 mM MgCl_2_, 0.3 μM solution of each primer, and 10–50 ng DNA template. HRM assays were performed using a LightCycler 480 II (Roche Diagnostics) according to the manufacturer’s instructions in 384-well plates. The annealing temperature was set at 60 °C. These strains and congenic lines were maintained at the Tokyo Metropolitan Institute of Medical Science (TMIMS) under specific-pathogen-free (SPF) conditions of temperature (23 ± 1 °C), relative humidity (50% ± 10%), and light/dark cycles (12 h/12 h), with food and water provided ad libitum.

### 2.2. Hearing Tests

For ABR and DPOAE recordings, mice were anesthetized with an intraperitoneal injection of a mixture of three anesthetic agents (medetomidine, 0.75 mg/kg; midazolam, 4 mg/kg; butorphanol, 5 mg/kg), and their core body temperatures were maintained at 38 ± 0.5 °C using PhysioSuite (Kent Scientific Corporation, Torrington, CT, USA). The recordings were performed on the left ears of male mice to avoid the confounding effects of bilateral ears and sex. Sex based differences have been reported in the ABR thresholds and DPOAE amplitudes of the B6 strain [22,23,24]. ABRs were recorded using a tone pip stimulus of 4, 8, 16, and 32 kHz using the TDT System III (TDT, Alachua, FL, USA) and BioSigRP software (TDT). The ABR thresholds and peak-to-trough ABR wave I amplitudes were determined as previously described [6,25]. The DPOAE amplitudes (2*f*_1_-*f*_2_) at 8, 11.3, 16, 22.6, and 32 kHz were recorded using the ER10X Extended Bandwidth Research Probe System (Etymotic Research, Elk Grove Village, IL, USA) and EMAV Plus software (version 3.32) (Etymotic Research), as previously described [6]. Data for the ABR thresholds in B6, B6-*Cdh23*^+/+^, and MSM mice as well as for the DPOAE amplitudes in B6 and B6-*Cdh23*^+/+^ mice were obtained from our previous study [6] and combined with the newly obtained data from each mouse in the present study.

### 2.3. Hair Cell Counts

For cochlear hair cell counting, the mice were anesthetized with isoflurane. Under anesthesia, the inner ears of male mice were dissected and fixed with 4% paraformaldehyde for 2 h at room temperature (RT), followed by incubation for 2 h at 4 °C. Thereafter, the sections were decalcified in 10% EDTA for 20–24 h at 4 °C (B6 strain) or 16–20 h at 4 °C (MSM strain). The cochleae were excised from the inner ear, permeabilized in 0.25% Triton X-100 for 30–60 min, and then subjected to three 5-min washes in PBS. After blocking with 0.5% blocking reagent (Roche Molecular Biochemicals, Indianapolis, IN, USA) for 1 h at RT, the cochleae were incubated with an anti-myosin VI (MYO6) primary antibody (Proteus Biosciences Inc., Ramona, CA, USA; 25-6791, 5 μg/mL) diluted in Can Get Signal Immunostain B (TOYOBO, Osaka, Japan) and stained overnight at 4 °C. Subsequently, the cochleae were washed three times for 5 min in PBS and incubated with a mixture of Alexa Fluor 568- or Alexa Fluor 633-conjugated secondary antibody (Thermo Fisher Scientific, Grand Island, NY, USA; A-11011 and A-21070, 5 μg/mL) and Alexa Fluor 488-conjugated phalloidin (Thermo Fisher Scientific; A12379, 4 units/mL) diluted in Can Get Signal Immunostain B (TOYOBO) for 1 h at RT. After three PBS washes, the cochleae were mounted onto glass slides using PermaFluor Aqueous Mounting Medium (Thermo Fisher Scientific) or ProLong Glass Antifade Mountant (Thermo Fisher Scientific). Fluorescence images were obtained using a Zeiss LSM 710 confocal microscope (Carl Zeiss, Jena, Germany), and images were processed using ZEN 2009 software (Carl Zeiss). The MYO6- and phalloidin-positive inner hair cells (IHCs) and outer hair cells (OHCs) were counted manually from 4 kHz (0.4 ± 0.07 mm from the apex), 8 kHz (1.0 ± 0.1 mm), 16 kHz (2.0 ± 0.1 mm), and 32 kHz (4.3 ± 0.2 mm) stimuli in the cochleae. 

### 2.4. Scanning Electron Microscopy

Phenotypic analysis of the cochlear stereocilia of male MSM mice at 1 and 24 months of age was performed using scanning electron microscopy (SEM), as previously described [25]. Stereociliary phenotypes were observed along the length of the cochlea corresponding to the 16 kHz sound stimulus. The cochlear specimens were examined using a Hitachi S-4800 field emission SEM instrument (Hitachi High-Technologies Corporation, Tokyo, Japan) at an accelerating voltage of 10 kV.

### 2.5. Linkage Analysis

PCR templates for linkage analysis were extracted from the pinna of the male (B6 × B6-Chr12C^MSM^) F_2_ progeny, as described above. The SSLP and SNP markers used for linkage analysis are listed in Appendix A, respectively. SNP genotyping for linkage analysis was performed as described previously [15] using the MassARRAY iPLEX Gold Assay (Agena Bioscience, Inc., San Diego, CA, USA). ABR thresholds at 32 kHz in 4-month-old mice were evaluated as traits for linkage analysis. The ABR thresholds were classified into two distinct groups: the “normal hearing” (NH) and “hearing loss” (HL). We categorized the F_2_ mice based on ABR thresholds into three groups (group A: NH, <35 dB and HL, >70 dB; group B: NH, <40 dB and HL, >60 dB; and group C: NH, <45 dB and HL, >50 dB) and performed three distinct linkage analyses between genotypes and phenotypes of each category group. The significant linkage between the three genotypes (B6/B6 homozygote, B6/MSM heterozygote, and MSM/MSM homozygote) and two phenotypes (NH and HL) was calculated by Chi-square (χ^2^) tests using GraphPad Prism 9 (GraphPad, San Diego, CA, USA). 

### 2.6. Statistical Analysis

Data of the ABR thresholds, DPOAE amplitudes, and hair cell survival rates are presented as mean ± standard deviation (SD) in line graphs. Data on the number of hair cells are presented as scatter plots with mean and SD. Differences in these data were analyzed using two-way factorial analysis of variance (ANOVA) with Sidak’s or Tukey’s multiple comparison test. Comparison of the ABR thresholds at 32 kHz among B6, (B6 × B6-Chr12C^MSM^) F_1_, and B6-Chr12C^MSM^ mice are presented as violin plots, with median and the 25th and 75th quartiles. Differences were analyzed using a one-way factorial ANOVA with Tukey’s multiple comparison test. The fitting curves for Gaussian distribution are depicted and superimposed over the histogram for the frequency distribution of ABR thresholds in (B6 × B6-Chr12C^MSM^) F_2_ mice. GraphPad Prism 9 software (GraphPad) was used to create graphs and calculate column statistics and *p*-values.

## 3. Results

### 3.1. Characterization of the ARHL Associated Phenotypes in MSM Strain

Although we previously reported that the hearing abilities of MSM strain are maintained over long durations, the evidence is preliminary because it is based on ABR threshold data from a limited number of mice [6,21]. Therefore, we first recorded the ABR thresholds of the MSM strain with numerous mice of ages 1, 2, 4, 8, 12, 16, 20, and 24 months and compared their age-related changes with those in the B6 strain.

A significant elevation in ABR threshold, assessed by post-hoc test, was first detected in the B6 strain at 4 months of age (Figure 1A and Appendix A) and was presented by the elevation in ABR thresholds at 32 kHz, as previously described [6] (Appendix A). The mean ABR threshold at 32 kHz reached levels indicative of severe (>70 dB SPL) and profound hearing loss (>90 dB SPL) at 8 and 12 months of age, respectively. With aging, the elevation in the ABR threshold in the B6 strain spread toward areas corresponding to 16, 8, and 4 kHz stimuli. The mean difference at all frequencies from the age of 1 month was 42.2 ± 16.1 dB SPL in the B6 strain at the end of 12 months of age. At 16 months of age, the mean ABR thresholds at all frequencies except 8 kHz reached levels indicative of severe hearing loss. Moreover, the mean ABR thresholds reached levels indicative of profound hearing loss at 20 months of age in response to the 4 and 16 kHz stimuli and at 24 months of age in response to the 8 kHz stimulus (Figure 1A). The mean ABR thresholds at all frequencies increased to 96.4 ± 6.0 dB SPL in the B6 strain at 24 months of age.

Age-related elevations in ABR thresholds were also observed in MSM mice (Figure 1A). However, there was a clear difference in the elevation patterns between the B6 and MSM strains. The elevation began with ABR thresholds at 4 kHz in the MSM strain. Highly significant age-related effects (*p* = 0.0001) in the ABR thresholds were detected in the MSM strain at 8 months of age, and the significance by post-hoc test was presented by the elevation in ABR thresholds at 4 kHz (Appendix A). After eight months of age, significant increases in ABR thresholds due to aging were detected in MSM strain. This significance is attributed not only to the elevation in the ABR thresholds at 4 kHz but also to those at 8, 16, and 32 kHz. However, the elevations in the ABR thresholds due to aging in the MSM strain were significantly lower than those in the B6 strain. The mean difference at all frequencies from 1 month of age was 11.8 ± 10.5 dB SPL in the MSM strain at 24 months of age and was approximately five times lower than that in the B6 strain.

Next, we recorded the DPOAE amplitudes of the MSM strain at 1, 2, 4, 8, 12, 16, 20, and 24 months of age and compared them with the age-related changes in the B6 strain. The DPOAE amplitudes gradually decreased from the high *f*_2_ frequency in the B6 strain with aging, as previously reported [6]. A highly significant effect of age on DPOAE amplitudes was first detected at 8 months of age, caused by a decrease in amplitudes at 22.6 and 32 kHz (Figure 1B and Appendix A). The amplitude at 16 kHz in the B6 strain exhibited a highly significant decrease at 12 months of age. At 16 months of age, the DPOAE amplitudes at all frequencies decreased to the noise floor levels in the B6 strain.

In contrast, the DPOAE amplitudes were stably detectable in the MSM strain until 24 months of age (Figure 1B). Although a significant difference (*p* < 0.01) in the DPOAE amplitudes between 1 and 24 months of age was detected in the MSM strain (Appendix A), the mean difference at all frequencies from 1 month of age was only 3.2 ± 5.8 dB SPL in MSM strain at 24 months of age.

Furthermore, we investigated the cochlear phenotypes of B6 and MSM strains at 2, 10, 18, and 24 months of age by counting IHCs and OHCs in the areas corresponding to the 4, 8, 16, and 32 kHz sound stimuli after immunostaining with phalloidin and anti-MYO6 antibody (Figure 2A,B). There were significant differences between the number of hair cells in the B6 and MSM strains at 2 months of age. More IHCs and OHCs were present in the MSM strain than in the B6 strain (Figure 2C). 

The number of IHCs of the B6 strain at 10 months of age were significantly decreased in the area corresponding to high-frequencies (Figure 3A,B, and Appendix A). The survival rates of IHCs at 32 and 16 kHz-corresponding areas at 10 months of age were 65.8 ± 18.7% and 68.5 ± 19.2%, respectively. Although the IHC survival rates between B6 mice of 10 and 18 months of age were approximately the same in all cochlear areas, a significant decrease in survival rate was detected in all areas at 24 months of age. The IHC survival rates at 4, 8, 16, and 32 kHz-corresponding areas at 24 months of age were 80.0% ± 20.7%, 63.6% ± 12.9%, 57.5% ± 11.6%, and 51.0% ± 6.9%, respectively. A highly significant decrease in OHC survival rates was also detected in the B6 strain at 10 months of age, caused by OHC loss at 4 and 32 kHz (Figure 3A,B and Appendix A). The decrease in the OHC survival rates in the 4 and 32 kHz-corresponding areas accelerated at 18 months. In particular, the OHC survival rates decreased to 11.4% ± 18.9% at the 32 kHz-corresponding area. At this age, a highly significant decrease was also detected in the 8 kHz-corresponding area in the B6 strain. At 24 months of age, most of the OHCs disappeared from all areas visualized for counting (Figure 3A,B). The OHC survival rates at 4, 8, and 16 kHz-corresponding areas were 14.0% ± 9.1%, 15.1% ± 11.7%, and 0.5% ± 1.2%, respectively, in B6 strain at 24 months of age. The OHC signal was not observed at the 32 kHz-corresponding area (Figure 3A).

In contrast to age-related loss of IHCs in the B6 strain, the number of IHCs in the MSM strain was maintained in all observed areas over long durations. More than 95% of the IHCs survived in all cochlear areas of the MSM strain at 24 months of age. The survival patterns of OHCs in the cochlear areas in the MSM strain were markedly different from those in the B6 strain. The OHC survival rate decreased in the apical area of the cochlea in the MSM strain. The OHC loss pattern in this strain was similar to that observed in the CBA/CaJ strain, which is a mouse model with good hearing abilities [26]. Highly significant decrease in the number of OHCs was detected at 4 and 8 kHz in the MSM strain at 10 months of age (Figure 4A,B and Appendix A). The OHC survival rates at 4, 8, and 16 kHz at 24 months of age were 19.7 ± 14.6%, 25.1 ± 2.2%, and 56.2 ± 18.0%, respectively. However, OHC loss at 32 kHz was not observed until 24 months of age (Figure 4A). The OHC survival rate was 93.6 ± 4.9% at 32 kHz (Figure 4B).

Next, we observed the stereocilia bundles in the aged MSM strain using SEM. Figure 5 shows the bundle morphologies from the 16 kHz-corresponding cochlear areas in the MSM strain at 1 and 24 months of age. Missing and misshapen stereocilia bundles were observed in several OHCs of the MSM strain at 24 months of age. However, there are perfectly normal stereocilia bundles that were also present in several OHCs at 24 months of age compared with those at 1 month of age, and the ‘V’-shaped and staircase architectures were not significantly disrupted in OHCs with misshapen stereocilia bundles.

### 3.2. Hearing Assessments in B6 Strain by Substitution to MSM-Derived Chromosome 17

Figure 6A shows the chromosomal structure of the B6-Chr17^MSM^ consomic strain. Although the pair of chromosome 17 was completely substituted with the MSM-derived chromosome, the B6-Chr17^MSM^ strain carried the homozygous *Cdh23^ahl^* allele, which is responsible for ARHL onset in the B6 strain [4,6,9]. 

We previously reported that the elevation in ABR thresholds in the B6-Chr17^MSM^ strain was suppressed until 18 months of age [21]. However, the frequency of the sound stimulus for ABR recording was limited to only 10 kHz. In the current study, we determined the ABR thresholds in the B6-Chr17^MSM^ strain at 4, 8, 16, and 32 kHz sound stimuli, which have been commonly and routinely presented in many previous studies for ABR recordings. The thresholds were then compared with those of the B6, B6-*Cdh23*^+/+^, and MSM strains. The aging-related change pattern of ABR thresholds at 4 kHz in the B6-Chr17^MSM^ strain was almost the same as that in the B6 strain (Figure 6B). Significant effects of substitution to MSM-derived chromosome 17 in the B6 strain were detected on ABR thresholds at 8, 16, and 32 kHz, but the thresholds in the B6 strain were obviously higher than those in the MSM strain (Figure 6B and Appendix A). Age-related elevations in ABR thresholds at 8 and 16 kHz in the B6-Chr17^MSM^ strain were more suppressed compared to those in the B6 strain. The suppression effects of thresholds at 12 and 16 months of age at both frequencies were supported by post-hoc analysis results (Figure 6B). Moreover, the changing patterns of ABR thresholds in the B6-Chr17^MSM^ strain with age were similar to those in the B6-*Cdh23*^+/+^ strain, which was edited by the allele from *ahl* (c.753A) to wild-type (c.753G) in the genetic background of the B6 strain [6,10]. Although the ARHL suppression effect on ABR thresholds at 32 kHz in the B6-Chr17^MSM^ strain was slightly weaker than that in the B6-*Cdh23*^+/+^ strain, the elevation in the ABR thresholds was significantly suppressed at 4, 8, and 12 months of age.

The recordings of the DPOAE amplitudes also confirmed the suppressive effects on the B6-Chr17^MSM^ strain. Reduction in DPOAE amplitudes at 22.6 and 32 kHz was significantly suppressed in the B6-Chr17^MSM^ strain at 8 months of age (Figure 6C and Appendix A). Although the DPOAE amplitudes of the B6-Chr17^MSM^ strain significantly decreased at 12 months of age compared to the B6-*Cdh23*^+/+^ and MSM strains, highly significant differences between B6 and B6-Chr17^MSM^ strains were detected at this age. The suppression effects in the form of decrease in DPOAE amplitudes at 16, 22.6, and 32 kHz were supported by post-hoc test results (Figure 6C). 

### 3.3. Hearing Assessments in B6 Strain by Transferring of MSM-Derived ahl3 Genomic Regions

Our previous study reported that the *ahl3* locus, which contributes to ARHL resistance in the B6-Chr17^MSM^ strain, is mapped to an approximately 22.3 Mb region on chromosome 17 [21] (Figure 7A). Therefore, we produced *ahl3* congenic lines by breeding of B6 and B6-Chr17^MSM^ strains to confirm the suppression effects on ARHL in the *ahl3* locus. Two *ahl3* congenic strains, B6.MSM-*ahl3*/1Tmims (B6.MSM-*ahl3*/1) and B6.MSM-*ahl3*/2Tmims (B6.MSM-*ahl3*/2), were used in this study (Figure 7A). The transfer of distinct homozygous MSM-derived *ahl3* genomic regions was confirmed in both the congenic strains by SSLP and SNP genotyping (Appendix A). 

We determined the ABR thresholds of both congenic strains at 4, 8, 16, and 32 kHz and compared these thresholds with those of the B6 and B6-Chr17^MSM^ strains. The ABR thresholds of the B6.MSM-*ahl3*/1 strain at different ages at 4, 8, and 16 kHz were higher than those of the B6 strain (Figure 7B and Appendix A). The changing pattern of ABR thresholds at 32 kHz in the B6.MSM-*ahl3*/1 strain with aging was almost the same as that in the B6 strain (Figure 7B). The changing patterns of the ABR thresholds in the B6.MSM-*ahl3*/1 strain differed substantially from those in the B6-Chr17^MSM^ strain. Significant differences were not detected in the ABR thresholds at 4 and 8 kHz between the B6 and B6.MSM-*ahl3*/2 strains. A highly significant difference (*p* < 0.001) between the B6 and B6.MSM-*ahl3*/2 strains was detected in the ABR thresholds at 16 kHz. The significant suppression effects in age-related elevation of the ABR threshold at 16 months of age were supported by post-hoc tests (Figure 7B). No significant differences were observed between the ABR thresholds of the B6.MSM-*ahl3*/2 and B6-Chr17^MSM^ strains at 16 kHz (Appendix A). A highly significant difference (*p* < 0.0001) between the B6 and B6.MSM-*ahl3*/2 strains was also detected in the ABR threshold at 32 kHz, but the suppression effect on elevation in the thresholds in the B6.MSM-*ahl3*/2 strain was weaker than that in the B6-Chr17^MSM^ strain. The elevation in ABR thresholds at 32 kHz in the B6.MSM-*ahl3*/2 strain was significantly suppressed at 4 and 8 months of age.

The recordings of the DPOAE amplitudes confirmed ARHL resistance for high-frequency sounds in the B6.MSM-*ahl3*/2 strain. Significant differences between the B6 and B6.MSM-*ahl3*/2 strains were detected in the DPOAE amplitudes at 4 months of age (Figure 7C and Appendix A). The significance is caused by the suppression of the amplitude decrease at both 22.6 and 32 kHz. The suppressive effect in the B6.MSM-*ahl3*/2 strain at this age was stronger than that in the B6-Chr17^MSM^ strain (Appendix A). The suppression effect on the reduced DPOAE amplitude against the high *f_2_* frequency sounds in the B6.MSM-*ahl3*/2 strain was maintained at 8 months of age and was similar to that in the B6-Chr17^MSM^ strain (Figure 7C). However, the suppression effects on the DPOAE amplitudes disappeared in the B6.MSM-*ahl3*/2 strain at 12 months of age (Appendix A).

### 3.4. ARHL Resistance in B6 Strain by Transferring MSM-Derived Centromeric Region on Chromosome 12

To screen novel loci associated with ARHL resistance in MSM, we continued measuring the ABR thresholds in other B6-Chr#^MSM^ consomic strains. In this process, we identified that a B6-Chr12C^MSM^ subconsomic strain, in which an approximately 69.8 Mb genomic region from MSM chromosome 12 is transferred to the B6 strain [14] (Figure 8A), exhibits ARHL resistance. Although the suppressive effects on age-related elevations in ABR thresholds in the B6-Chr12C^MSM^ strain were weaker than those in the B6-Chr17^MSM^ strain (Appendix A), significant differences between B6 and B6-Chr12C^MSM^ strains were detected in the thresholds at all frequencies (Figure 8B and Appendix A). In particular, the elevation in the ABR threshold at 4 kHz in the B6-Chr12C^MSM^ strain was significantly suppressed from the levels of the B6 strain to that of the MSM strain. Moreover, a highly significant difference between the B6 and B6-Chr12C^MSM^ strains was detected in ABR thresholds at 32 kHz (Appendix A). The early-onset elevation in ABR thresholds in the B6 strain at 32 kHz was also significantly suppressed in the B6-Chr12C^MSM^ strain (Figure 8B). 

### 3.5. Identification of the ahl10 Locus on Mouse Chromosome 12

To define the susceptibility and resistance loci underlying ARHL in B6 and MSM strains, we produced (B6 × B6-Chr12C^MSM^) F_1_ mice (*n* = 20) and measured the ABR thresholds at 32 kHz at 4 months, as this range had the highest mean difference in comparison to all the other frequencies and ages (Figure 8B). The number of B6 and B6-Chr12C^MSM^ mice at 4 months of age with ABR thresholds at 32 kHz was increased to 20 and 18, respectively, by additional measurements to increase the efficiency of data acquisition. The difference in mean thresholds between B6 and B6-Chr12C^MSM^ strains was 22.8 ± 23.1, with significance at *p* <0.05. The mean and distribution patterns of ABR thresholds in (B6 and B6-Chr12C^MSM^) F_1_ mice were similar to those in the B6-Chr12C^MSM^ strain (Figure 9A). The mean difference in thresholds between (B6 and B6-Chr12C^MSM^) F_1_ and B6-Chr12C^MSM^ mice was only 1.6 ± 21.7, whereas that between (B6 and B6-Chr12C^MSM^) F_1_ and B6 mice was 21.2 ± 24.6. Although the phenotype of (B6 and B6-Chr12C^MSM^) F_1_ strain suggests that a recessive susceptible allele on chromosome 12 in the B6 strain may be associated with elevation in ABR thresholds at 32 kHz, the presence of an MSM-derived dominant resistance allele was also predicted. Therefore, we performed intercrossing of (B6 and B6-Chr12C^MSM^) F_1_ mice, rather than backcrossing, to conduct linkage analysis of the loci associated with susceptibility or resistance in the elevation in ABR thresholds at 32 kHz. The ABR thresholds among (B6 and B6-Chr12C^MSM^) F_2_ mice at 4 months of age followed a bell-shaped Gaussian distribution (Figure 9B), suggesting the likelihood of contribution from multiple loci. 

To identify a novel locus associated with ARHL we performed linkage analysis using (B6 × B6-Chr12C^MSM^) F_2_ mice. For linkage analysis, the phenotypes of the F_2_ mice were categorized into three distinct groups (Figure 9C) based on ABR thresholds at 32 kHz sound stimuli, after which the linkage between genotypes and phenotypes was separately analyzed. In all groups, significant linkage associations (*p* = 0.01) were detected in markers on the centromeric region on chromosome 12 (Figure 9C). The F_2_ mice in the category B group showed a highly significant linkage (*p =* 0.001), with a peak at marker *rs50764161* (*p* < 0.001) located at a distance of 3.3 Mb from the centromere. The marker *rs36622082*, located ~6.9 Mb from the centromere, also showed strong correlation with ARHL susceptibility and resistance in all the three phenotype categories (*p* < 0.01, 0.001, and 0.01). In mice, the loci in the centromeric region of chromosome 12 that is responsible for ARHL are unknown; therefore, this locus was designated *ahl10* (Figure 9C).

To confirm the contribution of the *ahl10* locus in ARHL resistance in the MSM strain, we established the B6.MSM-*ahl10*/Tmims (B6.MSM-*ahl10*) congenic strain by crossing (B6 × B6-Chr12C^MSM^) F_2_ mice. The congenic strain was transferred to the MSM-derived 10.1Mb region within the *ahl10* candidate interval (Figure 10A). The ABR thresholds of the B6.MSM-*ahl10* strain were measured at 2, 4, 8, and 12 months of age and were deemed highly significant at all frequencies compared to the B6 strain (Figure 10B and Appendix A). The line graphs representing the ABR thresholds of the B6.MSM-*ahl10* and B6-Chr12C^MSM^ strains closely resembled each other at all frequencies (Figure 10B). There were no significant differences in the ABR thresholds between the two strains (Appendix A). However, significant differences were detected in the DPOAE amplitudes between the B6 and B6.MSM-*ahl10* strains. At 4 months of age, the decrease in the DPOAE amplitudes at 22.6 and 32 kHz was clearly suppressed in the B6.MSM-*ahl10* strain (Figure 10C). The decrease in the DPOAE amplitudes was suppressed at all frequencies except at 8 kHz in the B6.MSM-*ahl10* strain at 8 months of age; however, post-hoc analysis did not reveal any significant difference between B6 and B6.MSM-*ahl10* strains on frequency comparison (Figure 10C and Appendix A). 

### 3.6. Exploration of Candidate Genes in ahl3 and ahl10 Regions

We extracted 64,664 (57,165 SNPs, 3038 insertions, and 4461 deletions) and 48,295 (42,018 SNPs, 2626 insertions, and 3651 deletions) polymorphisms between B6 and MSM strains in B6.MSM-*ahl3*/2 and -*ahl10* congenic regions, respectively, using the MoG+ database [27]. The *ahl3*/2 and *ahl10* regions harbor 27 and 36 protein-coding genes, respectively. Although there were no insertions, deletions, or nonsense mutations in protein-coding genomic regions, we found non-synonymous mutations in B6 and MSM strains in each of the 21 genes within both the *ahl3*/2 and *ahl10* regions, as listed in Table 1 and Appendix A. 

In the *ahl3*/2 congenic region, 103 amino acid substitutions were detected in B6 and MSM strains. We validated the amino acid substitutions that allowed for potential phenotypic differences using SIFT [28], PROVEAN [29], and PolyPhen-2 [30] via VaProS [31]. Among the 103 amino acid substitutions, 20 were annotated as “Deleterious” or “Damaging” in the analysis using at least one software (Table 1). A total of 57 amino acid substitutions were identified between the *ahl10* regions of the B6 and MSM strains (Table 1 and Appendix A). Twelve substitutions were annotated as “Deleterious” or “Damaging” (Table 1). We explored whether the candidate genes listed in Table 1 were expressed in the neonatal cochlear epithelium [32] or adult IHCs/OHCs/supporting cells [33] in mice using the gEAR portal [34]. Both datasets revealed the expression of 18 genes in cochlear cells, except that of the thioredoxin domain containing 2 (*Txndc2*) and apolipoprotein B (*Apob*) genes.

## 4. Discussion

Genome-wide association studies and meta-analyses have been performed to identify genes causing susceptibility to ARHL in humans [35,36,37,38,39,40,41,42,43,44,45]. Although the causative genes and mutations remain mostly unknown, these studies identified many susceptible loci and high-risk SNPs associated with ARHL development and severity. However, little is known about the ARHL resistance genes and loci in humans. In mice, two studies [21,46] suggested the presence of loci, including *ahl3*, associated with ARHL resistance in the genetic background of inbred strains. Zheng et al. mapped the age-related hearing protection (*ahp*) locus in the 57–76 Mb region of chromosome 16 using the BXD strain, which is a recombinant inbred strain derived by crossing the B6 and DBA2 strains [46]. The *ahp* locus has strong ARHL resistance caused by the homozygosity of the *Cdh23^ahl^* allele. In this study, we suggest the roles of at least two resistance loci, *ahl3* and *ahl10*, underlying the genetic background of the MSM strain.

### 4.1. Pathological Factors That Prevent ARHL Onset in MSM Strain

ABR measurements confirmed that the threshold shifts due to aging were very small in the MSM strain (Figure 1A), indicating that the hearing ability of the MSM strain was stable for long durations. We predicted that a key factor responsible for the stability of hearing ability in MSM strain is the maintenance of OHC function until old age. Strong electrophysiological evidence suggests that DPOAE amplitudes can be stably detected in MSM strains at old age (Figure 1B). DPOAEs are sensitive indicators of OHC function [47,48]. The main function of OHCs is to support the detection of sounds with high sensitivity in the cochlea through the amplification of sound-induced vibrations by hair bundle motility and somatic motility [47,48,49]. Therefore, the stability of DPOAE amplitudes suggests that the OHCs of MSM strain maintain normally functioning hair bundles and/or somatic motilities at old age. However, the number of OHCs of the MSM strain decreased with age. Although there were no significant differences in DPOAE amplitudes at 8 and 16 kHz in MSM strain with age, the number of OHCs from the 8 and 16 kHz-corresponding areas decreased at 24 months of age (Figure 4B), indicating that the OHC loss at 8 and 16 kHz areas is not influenced by DPOAE amplitudes. Normal staircase patterns were observed in the stereocilia bundles of the remaining OHCs in the MSM strain at 24 months of age (Figure 5). These results suggest that the DPOAE amplitudes can be maintained by a smaller number of OHCs with normal stereocilia bundles.

We predicted that the prevention of IHC loss would be another key factor in maintaining auditory stability in the MSM strain. The number of IHCs did not decrease from any of the cochlear areas with age (Figure 4B). IHCs are conventional sensory receptors that transmit most acoustic information to the brain via ribbon synapses and type-I spiral ganglion neurons [49]. Therefore, IHC maintenance may be a crucial factor for ARHL prevention in MSM strain.

### 4.2. ARHL Resistance Effect in B6 Strain by Transferring MSM-Derived Chromosome 17 and ahl3 Genomic Regions 

We previously reported that the B6-Chr17^MSM^ strain exhibited ARHL resistance despite the presence of the *Cdh23^ahl^* allele [21]. In the present study, we confirmed ARHL resistance in the B6-Chr17^MSM^ consomic strain. In particular, strong resistance was detected at elevations in ABR thresholds at 8, 16, and 32 kHz (Figure 6B). The resistance effects in the ABR thresholds at 8 and 16 kHz were similar to those of mice with the wild-type *Cdh2*3^+^ allele edited from the *Cdh2**3^ahl^* allele. The resistance effect of ARHL was recognized by the decreasing DPOAE amplitudes against high-frequency sounds with aging (Figure 6C), suggesting that the substitution of MSM-derived chromosome 17 in the B6 strain prevents the decrease in OHC function. A delay in the elevation of ABR thresholds and a decrease in DPOAE amplitudes was detected in the B6.MSM-*ahl3*/2 congenic strain (Figure 7B,C) suggesting the presence of a genetic factor associated with ARHL resistance in the MSM strain. Although susceptibility loci and SNPs associated with ARHL and NIHL on chromosome 17 in inbred strains are reported [50,51,52], the genomic positions did not overlap with B6.MSM-*ahl3*/2 congenic region. 

Unfortunately, the information on heritability of ARHL resistance in the B6-Chr17^MSM^ consomic strain was missing, and the heritability decreased in several hearing phenotypes of the B6.MSM-*ahl3*/2 congenic strain. The resistance effect disappeared with an increase in ABR threshold at 8 kHz. The resistance effect of ABR thresholds at 32 kHz was weakened in the B6.MSM-*ahl3*/2 strain. Although the DPOAE amplitudes of the B6.MSM-*ahl3*/2 strain were similar to those of the B6-Chr17^MSM^ strain at 4 and 8 months of age, a significant difference between the amplitudes of the B6-Chr17^MSM^ and B6.MSM-*ahl3*/2 strains was detected at 12 months of age (Appendix A). These results indicate that one or more other genetic factor(s) are involved in a region distinct from the *ahl3* locus on chromosome 17. In a previous study, we mapped the *ahl3* locus using ABR thresholds at 10 kHz only. The mapping strategy can be considered a reason for missing other genetic factor(s) associated with ARHL resistance in the MSM strain.

In addition, genetic incompatibility by substitution on chromosome 17 may cause missing and decreasing heritability in the B6.MSM-*ahl3*/2 strain. Oka et al. conducted whole-genome transcriptional profiling in the B6-ChrX^MSM^ consomic strain and confirmed that approximately 20% of genes located in MSM-derived X-chromosomal regions were differentially expressed in the B6-ChrX^MSM^ strain compared to the B6 strain [53]. Moreover, substitution of MSM-derived chromosome X in the B6 strain led to the differential expression of many autosomal genes. Since the majority of the B6 strain genome is derived from *M. m. domesticus*, the results indicate that genome-wide differential gene expression occurred because of genetic incompatibility, potentially attributed to *M. m**. molossinus*-derived chromosomal substitution. In our case, ARHL resistance in the B6-Chr17^MSM^ strain may be related to a change in the expression profile due to the substitution of chromosome 17. Therefore, it is possible that the differential expression of several genes may be reversed by narrowing the MSM-derived genomic region of the B6.MSM-*ahl3*/2 strain, which may also lead to missing and decreasing heritability of the hearing phenotypes from the B6-Chr17^MSM^ strain.

### 4.3. ARHL Resistance Effect in B6 Strain by Transfer of MSM-Derived Chromosome 12C and ahl10 Genomic Regions

In this study, we confirmed that the B6-Chr12C^MSM^ subconsomic strain exhibits ARHL resistance. Resistance effects were detected in the ABR thresholds at all frequencies (Figure 8). Although the resistance effects on the elevation in the ABR thresholds at 16 and 32 kHz in the B6-Chr12C^MSM^ strain were smaller than those in the B6-Chr17^MSM^ strain, a significant resistance effect at 4 kHz was observed in the B6-Chr12C^MSM^ strain (Appendix A). Moreover, the novel *ahl10* locus was mapped to the centromeric 10.1 Mb region on chromosome 12 via linkage analysis and further phenotyping of the B6.MSM-*ahl10* congenic strain (Figure 9 and Figure 10). Although a quantitative trait locus associated with noise-induced endocochlear potential differences between the B6 and BALB/cJ strains was mapped on chromosome 12, the genomic positions did not overlap with the *ahl10*^MSM^ congenic region [54]. Therefore, we recommend *ahl10* as a novel locus associated with ARHL. 

The resistance effects on the elevation of ABR thresholds in the B6-Chr12C^MSM^ strain were strongly influenced by the MSM-*ahl10* congenic region, suggesting that *ahl10* is responsible for ARHL resistance in the B6-Chr12C^MSM^ strain. The resistance effect of the *ahl10* locus appears to play a role in delaying ARHL, notably at high frequencies during early stages of aging. The elevation of the ABR thresholds at 32 kHz was clearly suppressed in the B6.MSM-*ahl10* strain compared to that in the B6 strain at 4 months of age (Figure 10B). Furthermore, suppression of the decrease in the DPOAE amplitudes against 22.6 and 32 kHz frequencies was observed in the B6.MSM-*ahl10* strain at 4 months of age (Figure 10C). 

### 4.4. Prediction of Candidate Genes for ARHL Resistance by ahl3 and ahl10 loci

Among the *ahl3* candidate genes carrying non-synonymous SNPs listed in Table 1, we found that the gene encoding protein tyrosine phosphatase, receptor type, M (PTPRM) protein appeared to be expressed specifically in Deiters’ cells, through a search using the gEAR portal. PTPRM is a member of the protein tyrosine phosphatase multifunctional family that functions in different cellular pathways, including cell division, apoptosis, and cell differentiation [55]. Another member of this family, protein tyrosine phosphatase, receptor type, Q gene (*PTPRQ*), is responsible for the dominant form (DFNA73) and recessive form (DFNB84) of nonsyndromic hearing loss in humans [56,57]. Homozygous *Ptprq*-knockout (KO) mice developed stereocilia fusion of IHCs and stereocilia loss of OHCs [58]. Although phenotypes associated with hearing loss have not been reported in *Ptprm*-KO mice, these phenotypes may be worth investigating. 

Adenylate cyclase 3 gene (*Adcy3*) may be a candidate for the *ahl10* locus. This gene encodes a member of the adenylyl cyclase (ADCY) protein family. Previous studies have suggested that ADCY regulates signal transduction in the inner ear, such as mechanotransduction channel activity, by catalyzing the production of cyclic adenosine monophosphate from adenosine triphosphate [59,60]. Another ADCY family member, *ADCY1* (adenylate cyclase 1), is the causative gene for human non-syndromic recessive hearing loss, DFNB44 [61]. In zebrafish, the *Adcy1* mutants exhibited hair cell dysfunction [61], and ADCY1 and ADCY3 are classified into the same group based on their signaling properties [62]. Although phenotypes associated with hearing have not been reported in humans and mice, we might be able to study the amino acid substitution of ADCY3 as a candidate for ARHL resistance at the *ahl10* locus. 

There may exist other possible candidate genes responsible for ARHL resistance within the *ahl3* and *ahl10* loci. According to the data from the gEAR portal, the genes encoding lysosomal-associated protein transmembrane 4A gene (*Laptm4a*) and vesicle-associated membrane protein, associated protein A gene (*Vapa*), located within the *ahl3* locus, are highly and uniformly expressed in all cell types of the cochlea. Moreover, Currall et al. reported that the loss of lipid droplet associated hydrolase (LDAH) protein within *ahl10* locus is associated with sensorineural hearing loss in humans and ARHL in mice [63]. The ABR and DPOAE thresholds were significantly elevated in *Ldah*-KO male mice at 12 months of age. In addition, the International Mouse Phenotyping Consortium (http://www.mousephenotype.org/, accessed on 21 July, 2022) reported abnormal ABRs in KO mice of the Fer (fms/fps related) protein kinase gene (*Fer*) gene within the *ahl3* locus; and RAB10, member RAS oncogene family gene (*Rab10*) and ATPase family, AAA domain containing 2B gene (*Atad2b*) within the *ahl10* locus. There were no non-synonymous mutations between the B6 and MSM strains within the protein-coding regions of *Ldah*, *Fer*, and *Rab10* genes. However, these genes cannot be excluded from being considered as candidate genes within the *ahl3* and *ahl10* loci because there is a possibility that polymorphisms in the regulatory regions of gene expression are associated with ARHL resistance. 

Finally, we suggest that there is a possibility that ARHL in the B6 strain was suppressed by genetic incompatibility caused by the transfer of subspecies-derived genomic regions, as reported previously during modifications of several phenotypes [53,64,65]. Even though the transferred genomic regions were not very long, we cannot completely deny the genetic effects of substitutions in the genome between the subspecies. 

## 5. Conclusions

Our data suggest that the *ahl3* and *ahl10* loci contribute to developmental delay of ARHL in the B6 strain and ARHL resistance in the MSM strain. The delay of ARHL in the B6 strain by both loci may suppress the effects of susceptible alleles underlying the delay in the B6 strain. However, we predicted that the genetic factors in both loci exerted ARHL-resistant effects, as congenic strains of both loci exhibited ARHL resistance despite having the homozygous *Cdh23^ahl^* allele. Therefore, we predict that causative genes and genomic polymorphisms of the *ahl3* and *ahl10* loci will become new targets for the prevention and treatment of ARHL. The MSM strain is a wild-derived inbred strain, and as most wild-derived inbred strains exhibit good hearing [2,4,5], we expect that wild-derived inbred strains can be powerful bioresources for the identification of genetic factors and elucidation of molecular mechanisms associated with ARHL resistance.

## Figures and Tables

**Figure 1 biomedicines-10-02221-f001:**
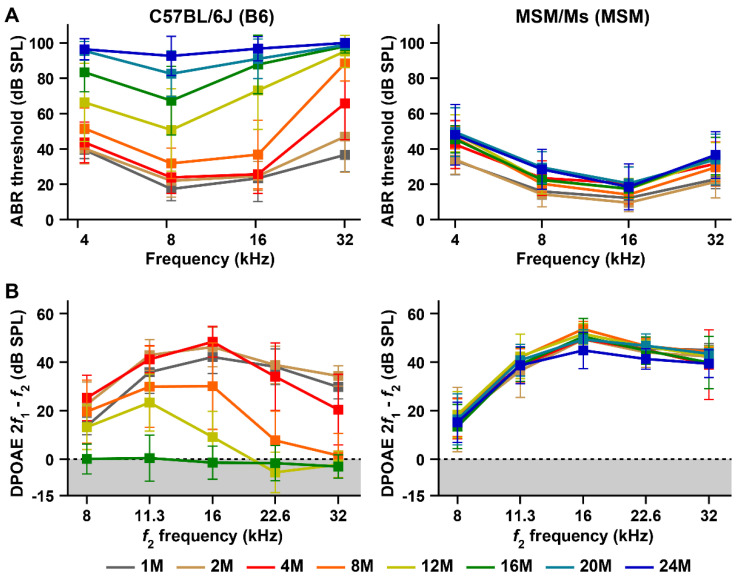
Characterization of age-related hearing loss (ARHL) in C57BL/6J (B6) and MSM/Ms (MSM) strains by measurements of the ABR thresholds and DPOAE amplitudes: (**A**) Comparisons of ABR thresholds among the ages (in months) in the B6 and MSM strains. ABR thresholds at 4, 8, 16, and 32 kHz sound stimuli were measured in male mice at 1 (B6, *n* = 10; MSM, *n* = 10), 2 (B6, *n* = 15; MSM, *n* = 12), 4 (B6, *n* = 15; MSM, *n* = 15), 8 (B6, *n* = 14; MSM, *n* = 11), 12 (B6, *n* = 15; MSM, *n* = 14), 16 (B6, *n* = 12; MSM, *n* = 11), 20 (B6, *n* = 16; MSM, *n* = 10), and 24 (B6, *n* = 10; MSM, *n* = 10) months of age. (**B**) Comparisons of DPOAE amplitudes in the B6 and MSM mice of different ages (in months). The DPOAE amplitudes (2*f*_1_-*f*_2_) against the *f*_2_ frequencies at 8, 11.3, 16, 22.6, and 32 kHz were recorded in male mice at 1 (B6, *n* = 10; MSM, *n* = 11), 2 (B6, *n* = 12; MSM, *n* = 15), 4 (B6, *n* = 15; MSM, *n* = 15), 8 (B6, *n* = 14; MSM, *n* = 12), 12 (B6, *n* = 15; MSM, *n* = 12), 16 (B6, *n* = 10; MSM, *n* = 10), 20 (MSM, *n* = 9), and 24 (MSM, *n* = 10) months of age. The means (squares) and SDs (error bars) of the ABR thresholds and DPOAE amplitudes are shown in each line graph. The noise floors (gray zones) are adjusted to 0 dB SPL.

**Figure 2 biomedicines-10-02221-f002:**
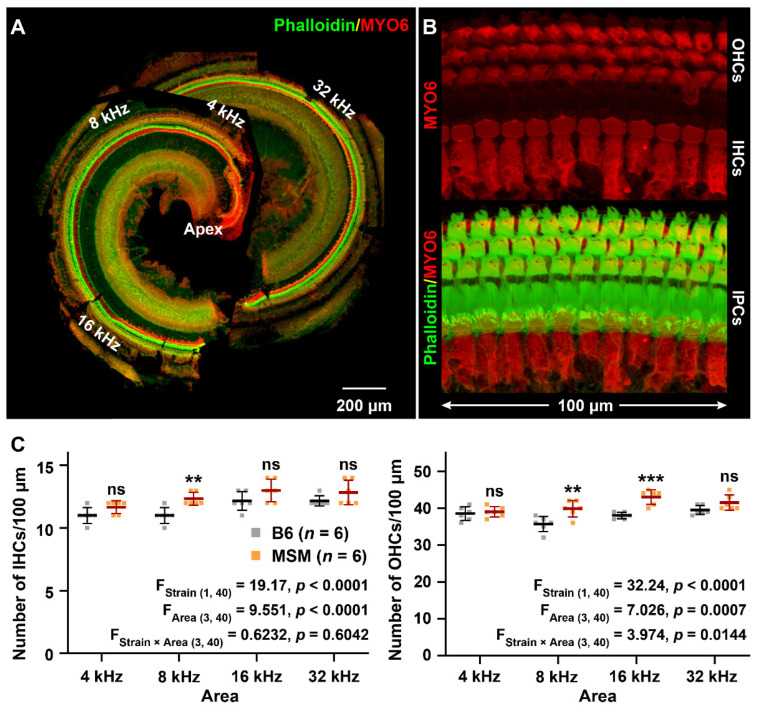
Counts of the inner hair cells (IHCs) and outer hair cells (OHCs) of B6 and MSM strains: (**A**,**B**) Confocal images of the cochlea and hair cells from the basal area in B6 strain at 2 months of age immunostained with anti-MYO6 antibody (red) and phalloidin (green). Cochlear hair cells were observed at the 4, 8, 16, and 32 kHz-corresponding areas and OHCs and IHCs were counted within 100 μm in this study. Inner pillar cells (IPCs). (**C**) Counts of the IHCs and OHCs within 4, 8, 16, and 32 kHz-corresponding areas of the cochlea from male mice of the B6 and MSM strains at 2 months of age. The means (middle bars) and SDs (error bars) of the number of IHCs and OHCs are shown in each graph. The square dots represent the number of cells of individual mice. The data used for the statistical analyses by two-way ANOVA are provided in the graphs for each strain and area (**B**). The asterisks and “ns” symbols indicate the presence of significant (** *p* < 0.01 and *** *p* < 0.001) and no significant differences, respectively, in the numbers of IHCs (left) and OHCs (right) between male B6 and MSM strains in the same area, determined using Sidak’s (IHCs) and Tukey’s (OHCs) multiple comparison tests after two-way ANOVA.

**Figure 3 biomedicines-10-02221-f003:**
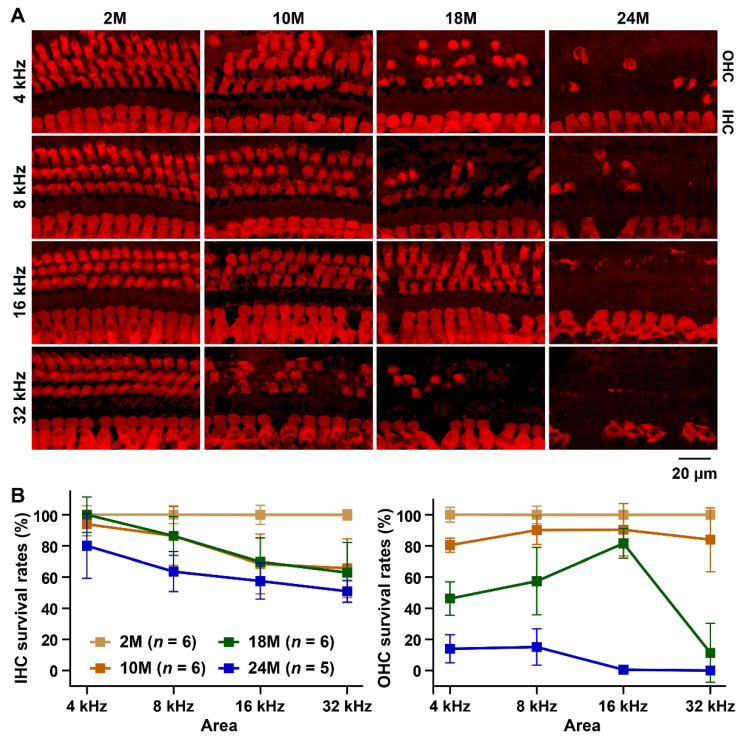
Decrease in the IHC and OHC survival in aging B6 strain (**A**) Representative confocal images of MYO6-positive IHC and OHC signals within the 4, 8, 16, and 32 kHz areas of the cochlea from male mice of the B6 strain at 2, 10, 18, and 24 months of age. (**B**) Percentage of surviving IHCs and OHCs at the 4, 8, 16, and 32 kHz-corresponding areas in the cochlea of male mice of the B6 strain at 2, 10, 18, and 24 months of age. The means (squares) and SD (error bars) of the IHC and OHC survivors are shown in each line graph. The mean numbers (Figure 2) of IHCs and OHCs from each area in the B6 mice at 2 months of age were assigned an arbitrary value of 100%.

**Figure 4 biomedicines-10-02221-f004:**
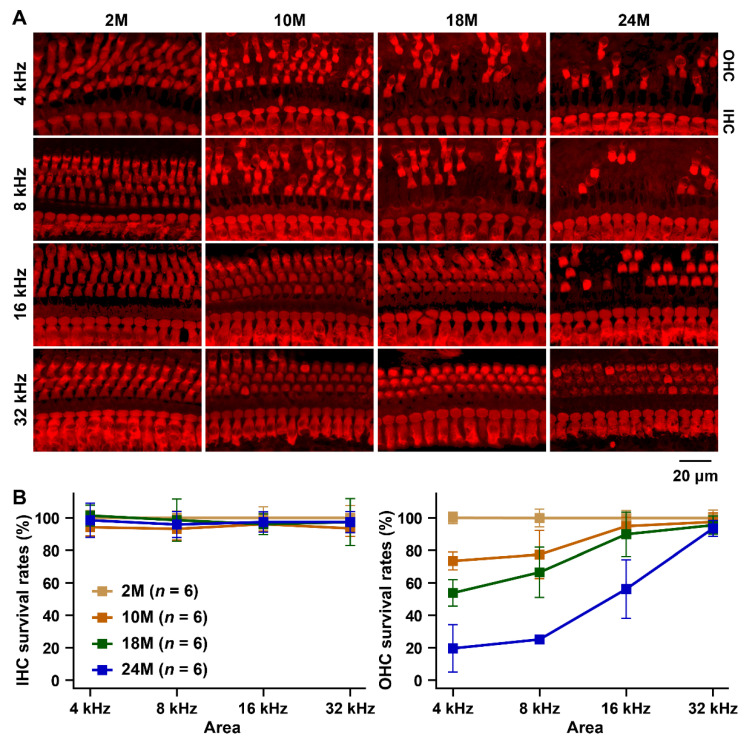
Decrease and/or maintenance in the IHC and OHC survival in aging MSM strain: (**A**) Representative confocal images of MYO6-positive IHCs and OHCs signals within the 4, 8, 16, 32 kHz areas of the cochlea from male mice of the MSM strain at 2, 10, 18, and 24 months of age. (**B**) Percentage of surviving IHCs and OHCs at the 4, 8, 16, 32 kHz areas in the cochlea of male mice of the MSM strain at 2, 10, 18, and 24 months of age. The means (squares) and SDs (error bars) of the IHC and OHC survivors are shown in each line graph. The mean numbers (Figure 2) of IHCs and OHCs from each area in the MSM strain at 2 months of age were assigned an arbitrary value of 100%.

**Figure 5 biomedicines-10-02221-f005:**
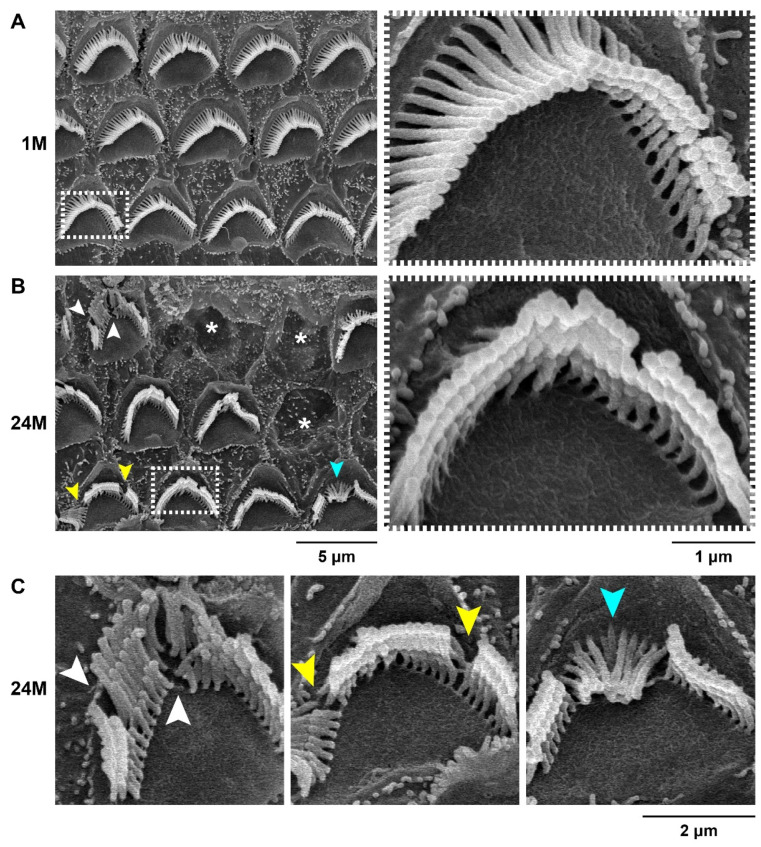
Maintenance and degeneration of the OHC stereocilia bundles in aging MSM strain: (**A**,**B**) SEM images showing stereocilia in the OHCs within the 16 kHz area of the male MSM mice at 1 and 24 months of age. The right panels are highly magnified images of the white dotted boxes shown in the left panels and show normal staircase patterns of the OHC stereocilia bundles. The asterisks indicate OHCs with absent bundles. (**C**) Abnormal OHC stereocilia bundles in MSM mice at 24 months of age. The images are magnified with OHCs pointed with arrowheads in the left panel in (**B**). The white, yellow, and cyan arrowheads indicate the absence of bundles, separation between the bundles, and inclined bundles, respectively.

**Figure 6 biomedicines-10-02221-f006:**
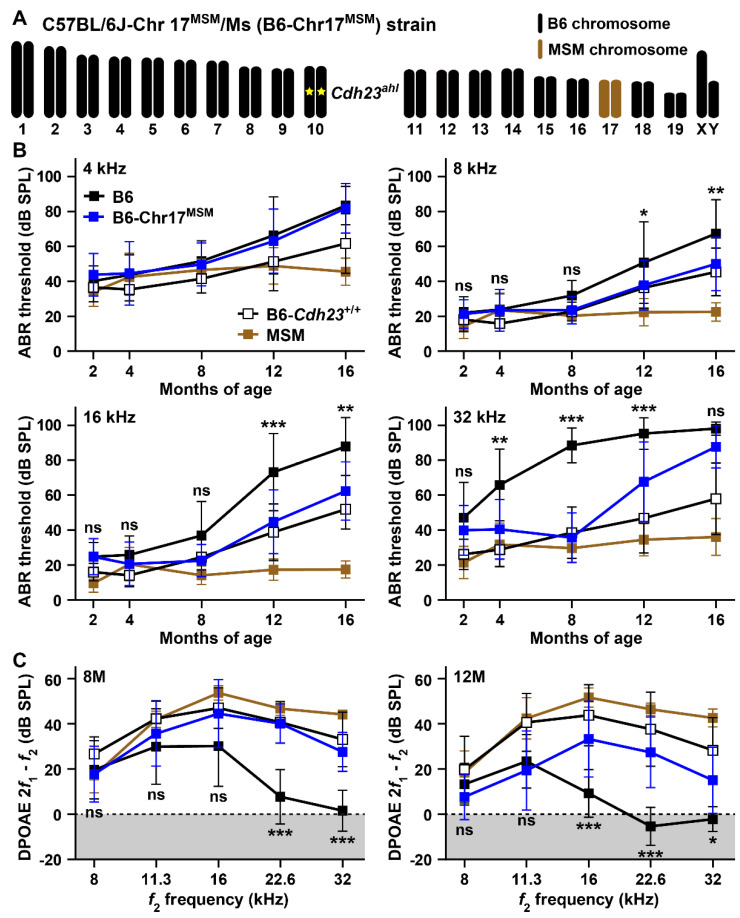
The C57BL/6J-Chr 17^MSM^/Ms (B6-Chr17^MSM^) consomic strain exhibits delayed ARHL onset: (**A**) Schematic chromosomal structure of a B6-Chr17^MSM^ strain. A yellow star on chromosome 10 indicates *Cdh23^ahl^* allele. (**B**) Comparison of the ABR thresholds at 4, 8, 16, and 32 kHz sound stimuli among male mice of the B6, MSM, B6-*Cdh23*^+/+^, and B6-Chr17^MSM^ strains at 2 (B6, *n* = 15; MSM, *n* = 12; B6-*Cdh23*^+/+^, *n* = 12; B6-Chr17^MSM^, *n* = 16), 4 (B6, *n* = 15; MSM, *n* = 15; B6-*Cdh23*^+/+^, *n* = 12; B6-Chr17^MSM^, *n* = 15), 8 (B6, *n* = 14; MSM, *n* = 11; B6-*Cdh23*^+/+^, *n* = 12; B6-Chr17^MSM^, *n* = 16), 12 (B6, *n* = 15; MSM, *n* = 14; B6-*Cdh23*^+/+^, *n* = 12; B6-Chr17^MSM^, *n* = 18), and 16 (B6, *n* = 12; MSM, *n* = 11; B6-*Cdh23*^+/+^, *n* = 10; B6-Chr17^MSM^, *n* = 12) months of age. (**C**) Comparison of DPOAE (2*f*_1_-*f*_2_) amplitudes against the *f*_2_ frequencies at 8, 11.3, 16, 22.6, and 32 kHz among male mice of the B6, MSM, B6-*Cdh23*^+/+^, and B6-Chr17^MSM^ strains at 8 (B6, *n* = 14; MSM, *n* = 12; B6-*Cdh23*^+/+^, *n* = 13; B6-Chr17^MSM^, *n* = 11) and 12 (B6, *n* = 15; MSM, *n* = 12; B6-*Cdh23*^+/+^, *n* = 12; B6-Chr17^MSM^, *n* = 12) months of age. The means (squares) and SDs (error bars) of the ABR thresholds and DPOAE amplitudes are shown. Two-way ANOVA revealed significant strain effects between B6 and B6-Chr17^MSM^ strains of the ABR thresholds at 8, 16, and 32 kHz and the DPOAE amplitudes at 8 and 12 months of age (Appendix A). The asterisks and “ns” symbols indicate significant (* *p* < 0.05, ** *p* < 0.01, and *** *p* < 0.001) and no significant differences, respectively, between B6 and B6-Chr17^MSM^ strains at the same age of the ABR thresholds and at the same frequency of the DPOAE amplitudes, determined using Sidak’s (8 kHz in B) and Tukey’s (16 and 32 kHz in (**B**,**C**)) multiple comparison tests after two-way ANOVA. The noise floors (gray zones) are adjusted to 0 dB SPL in (**C**).

**Figure 7 biomedicines-10-02221-f007:**
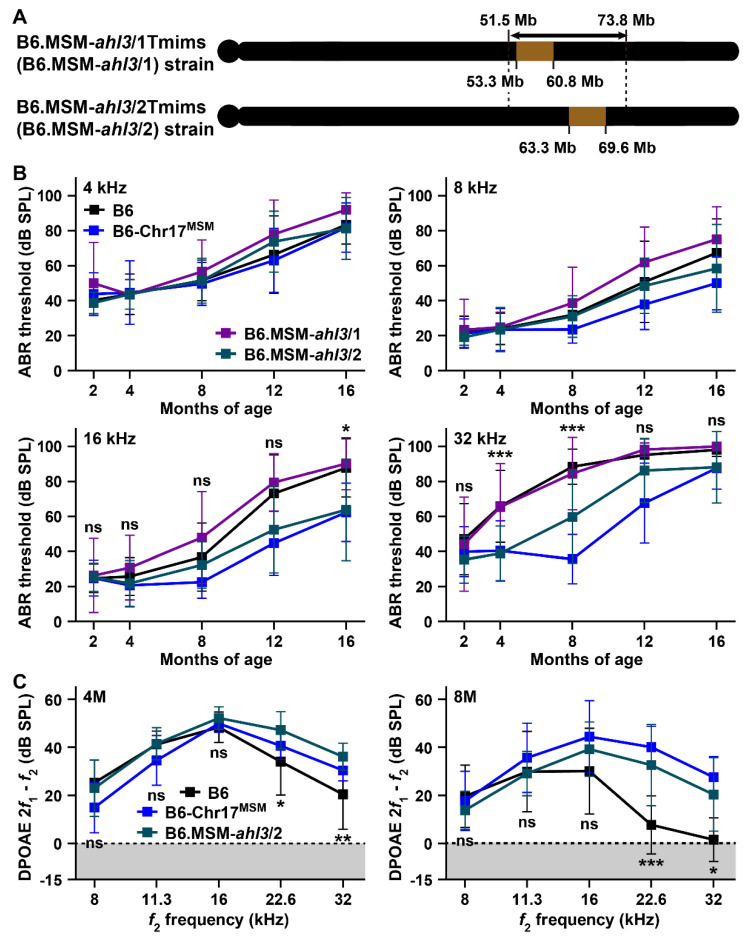
A B6.MSM-*ahl3* congenic line exhibits delayed ARHL onset at middle- and high-frequency sounds: (**A**) Chromosomal structure of the B6.MSM-*ahl3*/1Tmims (B6.MSM-*ahl3*/1) and B6.MSM-*ahl3*/2Tmims (B6.MSM-*ahl3*/2) congenic lines. Black and light brown colors indicate recipient B6 and transferred MSM chromosomal regions in both congenic lines, respectively. The interval of the double-headed arrow shows the candidate region of the *ahl3* locus estimated in the previous study [21]. (**B**) Comparison of the ABR thresholds at 4, 8, 16, and 32 kHz sound stimuli among male mice of the B6, B6-Chr17^MSM^, B6.MSM-*ahl3*/1, and B6.MSM-*ahl3*/2 strains at 2 (B6, *n* = 15; B6-Chr17^MSM^, *n* = 16; B6.MSM-*ahl3*/1, *n* = 12; B6.MSM-*ahl3*/2, *n* = 12), 4 (B6, *n* = 15; B6-Chr17^MSM^, *n* = 15; B6.MSM-*ahl3*/1, *n* = 14; B6.MSM-*ahl3*/2, *n* = 12), 8 (B6, *n* = 14; B6-Chr17^MSM^, *n* = 16; B6.MSM-*ahl3*/1, *n* = 11; B6.MSM-*ahl3*/2, *n* = 14), 12 (B6, *n* = 15; B6-Chr17^MSM^, *n* = 18; B6.MSM-*ahl3*/1, *n* = 11; B6.MSM-*ahl3*/2, *n* = 15), and 16 (B6, *n* = 12; B6-Chr17^MSM^, *n* = 12; B6.MSM-*ahl3*/1, *n* = 11; B6.MSM-*ahl3*/2, *n* = 11) months of age. (**C**) Comparison of DPOAE (2*f*_1_-*f*_2_) amplitudes against the *f*_2_ frequencies at 8, 11.3, 16, 22.6, and 32 kHz among male mice of the B6, B6-Chr17^MSM^, and B6.MSM-*ahl3*/2 strains at 8 (B6, *n* = 14; B6-Chr17^MSM^, *n* = 11; B6.MSM-*ahl3*/2, *n* = 10) and 12 (B6, *n* = 15; B6-Chr17^MSM^, *n* = 12; B6.MSM-*ahl3*/2, *n* = 12) months of age. The means (squares) and SDs (error bars) of the ABR thresholds and DPOAE amplitudes are shown. The noise floors (gray zones) are adjusted to 0 dB SPL. Two-way ANOVA revealed significant strain effects between the ABR thresholds of the B6 and B6-Chr17^MSM^ strains at 16 and 32 kHz and between the corresponding DPOAE amplitudes at 4 and 8 months of age (Appendix A). The asterisks and “ns” symbols indicate significant (* *p* < 0.05, ** *p* < 0.01, and *** *p* < 0.001) and no significant differences, respectively, between the ABR thresholds of the B6 and B6.MSM-*ahl3*/2 strains at the same age of the ABR thresholds and at the same frequency of the DPOAE amplitudes, determined using Sidak’s (32 kHz in (**B**)) and Tukey’s (16 kHz in (**B**); and 8 months (8M) in (**C**)) multiple comparison tests after two-way ANOVA. The noise floors (gray zones) are adjusted to 0 dB SPL in **C**.

**Figure 8 biomedicines-10-02221-f008:**
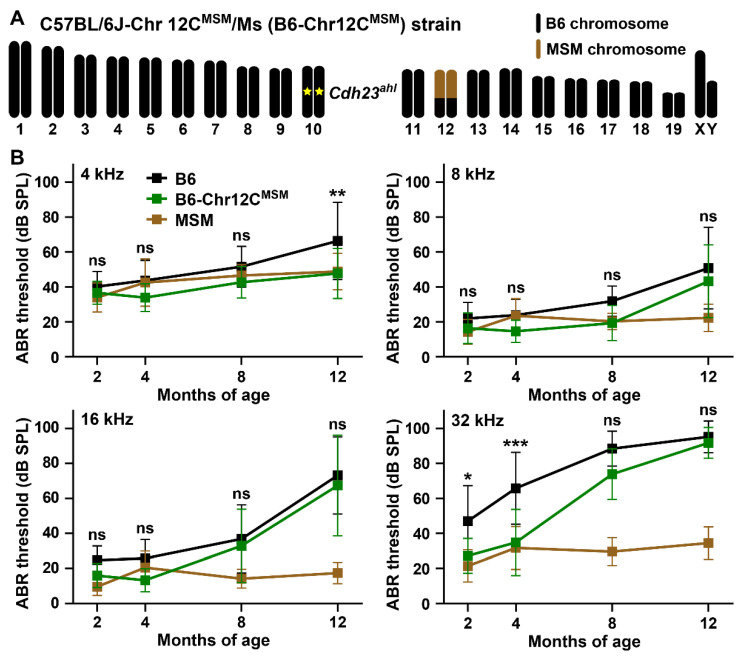
The C57BL/6J-Chr 12C^MSM^/Ms (B6-Chr12C^MSM^) subconsomic strain exhibits delayed ARHL onset: (**A**) Schematic chromosomal structure of a B6-Chr12C^MSM^ strain. A yellow star on chromosome 10 indicates *Cdh23^ahl^* allele. Light brown color indicates a transferred MSM-derived ~71 Mb chromosomal region. (**B**) Comparison of the ABR thresholds at 4, 8, 16, and 32 kHz sound stimuli among male mice of the B6, MSM, and B6-Chr12C^MSM^ strains at 2 (B6, *n* = 15; MSM, *n* = 12; B6-Chr12C^MSM^, *n* = 10), 4 (B6, *n* = 15; MSM, *n* = 15; B6-Chr12C^MSM^, *n* = 10), 8 (B6, *n* = 14; MSM, *n* = 11; B6-Chr12C^MSM^, *n* = 12), and 12 (B6, *n* = 15; MSM, *n* = 14; B6-Chr12C^MSM^, *n* = 11) months of age. The means (squares) and SDs (error bars) of the ABR thresholds are shown. Two-way ANOVA revealed significant strain effects between B6 and B6-Chr12C^MSM^ strains of the ABR thresholds at all frequencies (Appendix A). The asterisks and “ns” symbols indicate significant (* *p* < 0.05, ** *p* < 0.01, and *** *p* < 0.001) and no significant differences, respectively, between the ABR thresholds of the B6 and B6-Chr12^MSM^ strains at the same age in each graph, determined using Sidak’s (4, 8, and 16 kHz) and Tukey’s (32 kHz) multiple comparison tests after two-way ANOVA.

**Figure 9 biomedicines-10-02221-f009:**
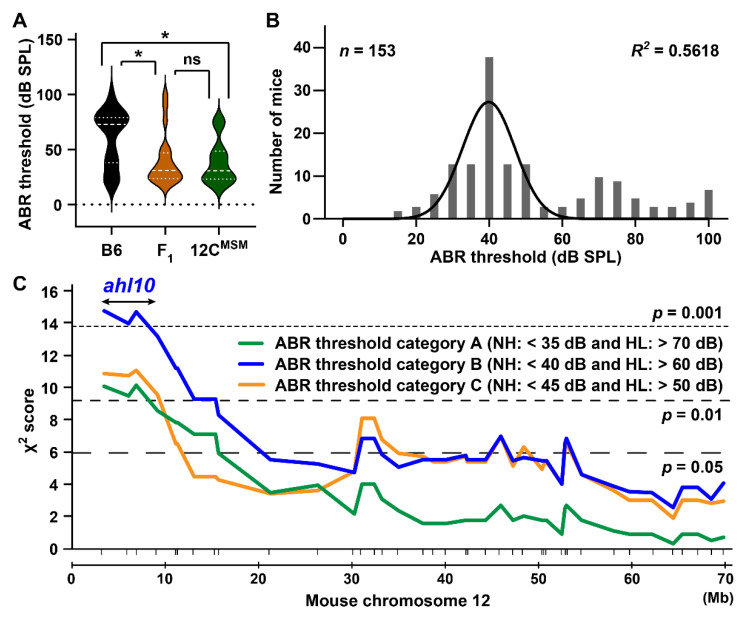
Identification of the genomic region (*ahl10* locus: double-headed arrow) associated with high-frequency hearing loss on chromosome 12: (**A**) Distributions of ABR thresholds at 32 kHz sound stimulus of the male B6 (*n* = 20), (B6 × B6-Chr12C^MSM^) F_1_ (*n* = 20), and B6-Chr12C^MSM^ (*n* = 18) mice at 4 months of age. The dashed and dotted lines within violin plots indicate the median and quartiles, respectively. Significant genotype effect (F_2, 55_ = 5.9, *p* = 0.0048) of the ABR thresholds were detected by one-way ANOVA analysis. Results of Tukey’s multiple comparison tests are given (* *p* < 0.05 and ns, no significant difference). (**B**) Distributions of ABR thresholds at 32 kHz stimulus among the (B6 × B6-Chr12C^MSM^) F_2_ mice at 4 months of age. The best-fit curves for a Gaussian distribution are shown. The goodness of fit for the curve is given as R squared (*R^2^*), calculated by the GraphPad software. (**C**) A linkage map of the *ahl10* locus (double-headed arrow), which is associated with resistance of the elevation of ABR thresholds at 32 kHz sound stimulus, using male (B6 × B6-Chr12C^MSM^) F_2_ mice at 4 months of age. The green (category A), blue (category B), and orange (category C) lines are χ^2^ score plots, which are calculated from three category groups based on classifications of the normal hearing (category A, *n* = 37; B, *n* = 56; C, *n* = 76) and hearing loss (category A, *n* = 39; B, *n* = 49; C, *n* = 64) of ABR thresholds at 32 kHz stimulus and three genotypes (B6/B6, B6/MSM, and MSM/MSM) in F_2_ mice. The marker positions (vertical bars; see Appendix A) are given below the linkage maps. The three horizontal dotted and dashed lines indicate statistically significant χ^2^ scores.

**Figure 10 biomedicines-10-02221-f010:**
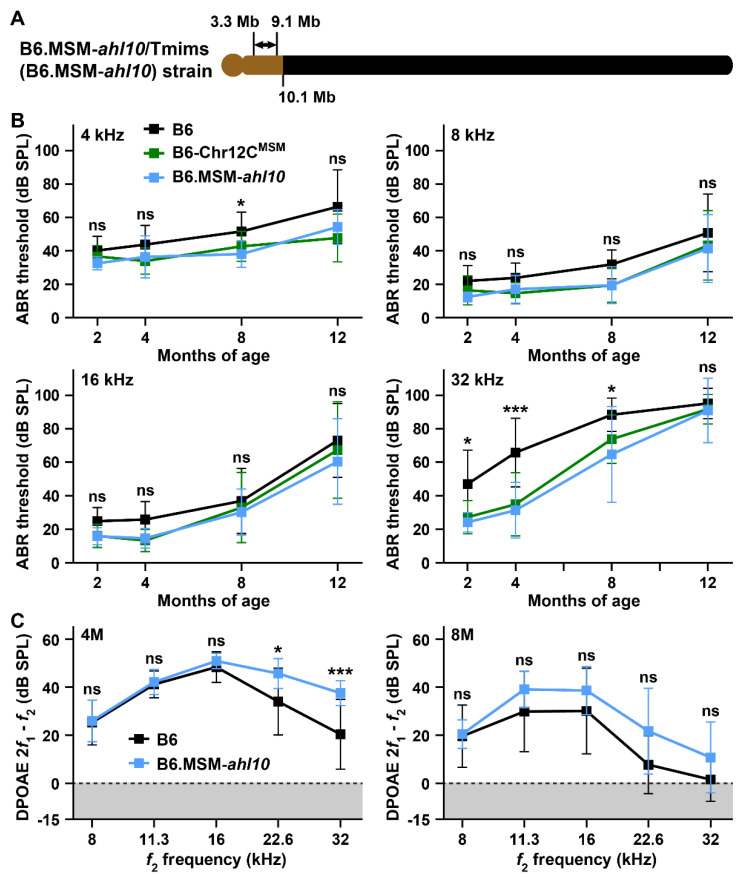
An MSM-*ahl10* congenic line exhibits significant suppressive effects of ARHL: (**A**) Chromosomal structure of B6.MSM-*ahl10*/Tmims (B6.MSM-*ahl10*) congenic line. Black and light brown colors indicate recipient B6 and transferred MSM chromosomal regions, respectively. The interval of the double-headed arrow shows the candidate region of the *ahl10* locus estimated by linkage analysis (Figure 9). (**B**) Comparison of the ABR thresholds at 4, 8, 16, and 32 kHz sound stimuli among male mice of the B6, B6-Chr12C^MSM^, and B6.MSM-*ahl10* strains at 2 (B6, *n* = 15; B6-Chr12C^MSM^, *n* = 10; B6.MSM-*ahl10*, *n* = 10), 4 (B6, *n* = 15; B6-Chr12C^MSM^, *n* = 10; B6.MSM-*ahl10*, *n* = 15), 8 (B6, *n* = 14; B6-Chr12C^MSM^, *n* = 12; B6.MSM-*ahl10*, *n* = 11), and 12 (B6, *n* = 15; B6-Chr12C^MSM^, *n* = 11; B6.MSM-*ahl10*, *n* = 10) months of age. (**C**) Comparison of DPOAE (2*f*_1_-*f*_2_) amplitudes against the *f*_2_ frequencies (8, 11.3, 16, 22.6, and 32 kHz) between male mice of the B6 and B6.MSM-*ahl10* strains at 4 (B6, *n* = 15; B6.MSM-*ahl10*, *n* = 15) and 8 (B6, *n* = 14; B6.MSM-*ahl10*, *n* = 11) months of age. The means (squares) and SDs (error bars) of the ABR thresholds and DPOAE amplitudes are shown. Two-way ANOVA revealed significant strain effects between B6 and B6.MSM-*ahl10* strains of the ABR thresholds at all frequencies and the DPOAE amplitudes at 4 and 8 months of age (Appendix A). The asterisks and “ns” symbols indicate significant (* *p* < 0.05 and *** *p* < 0.001) and no significant differences, respectively, between the ABR thresholds of the B6 and B6.MSM-*ahl10* strains at the same age in each graph, determined using Sidak’s (4, 8, and 16 kHz in (**B**); and 8 months (8M) in (**C**)) and Tukey’s (32 kHz in (**B**); and 4 months (4M) in (**C**)) multiple comparison tests after two-way ANOVA. The noise floors (gray zones) are adjusted to 0 dB SPL in (**C**).

**Table 1 biomedicines-10-02221-t001:** Amino acid substitutions that allow for potential phenotypic differences between B6 and MSM strains within B6.MSM-*ahl3*/2 and B6.MSM-*ahl10* congenic regions.

Congenic Region	Protein	Amino acid Substitution			Possible Impact		
		Position	B6	MSM	SIFT	PROVEAN	PolyPhen-2
*ahl3*/2	FBXL17	182	Pro	Ser	Deleterious	Neutral	Benign
	TMEM232	285	Thr	Asn	NA *	Deleterious	Possibly damaging
	TXNDC2	133	Glu	Gly	Tolerated	Deleterious	Benign
		203	Val	Ile	Tolerated	Neutral	Probably damaging
	ANKRD12	910	Asn	Ser	Deleterious	Neutral	Benign
	DDX11	657	Val	Met	Tolerated	Deleterious	Benign
	MTCL1	848	Arg	His	Tolerated	Deleterious	Benign
		1254	Ile	Leu	Deleterious	Neutral	Benign
	PTPRM	263	Cys	Arg	Tolerated	Deleterious	Probably damaging
	LAMA1	656	Asp	Asn	Tolerated	Deleterious	Benign
		1122	Gly	Ser	Tolerated	Deleterious	Benign
		1131	Ala	Val	Deleterious	Deleterious	Possibly damaging
		1994	Met	Thr	Tolerated	Deleterious	Possibly damaging
		2120	Val	Ala	Tolerated	Deleterious	Probably damaging
		3033	Ile	Val	Tolerated	Neutral	Possibly damaging
	ARHGAP28	256	Ala	Pro	Deleterious	Deleterious	Benign
	TMEM200C	249	Ser	Pro	Tolerated	Deleterious	Benign
		344	His	Asn	Deleterious	Neutral	Benign
		633	Lys	Glu	Deleterious	Deleterious	Probably damaging
	EPB41L3	803	Ile	Val	Deleterious	Neutral	Benign
*ahl10*	ASXL2	940	Ile	Thr	Tolerated	Deleterious	Possibly damaging
		1180	Gln	Lys	Tolerated	Neutral	Probably damaging
		1191	Ser	Phe	Tolerated	Deleterious	Probably damaging
	ADCY3	113	Phe	Val	Deleterious	Deleterious	Benign
	CENPO	177	Arg	Ser	Tolerated	Neutral	Probably damaging
	GM17541	46	Leu	Ser	Tolerated	Deleterious	Benign
		94	Gly	Glu	Deleterious	Neutral	Benign
	FAM228A	113	Gln	His	Tolerated	Deleterious	Benign
		267	Arg	Gly	Tolerated	Neutral	Probably damaging
	WDCP	474	Pro	Ala	Tolerated	Deleterious	Benign
	ATAD2B	702	Thr	Ile	Deleterious	Neutral	Benign
	APOB	260	Thr	Lys	Deleterious	Deleterious	Probably damaging
		459	Thr	Lys	Deleterious	Neutral	Probably damaging
	MATN3	333	Ala	Thr	Tolerated	Deleterious	Benign

* NA, Not annotated.

## Data Availability

All the data presented in this study are available from the corresponding author upon reasonable request.

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
