# Peer review of "Two Loci Contribute to Age-Related Hearing Loss Resistance in the Japanese Wild-Derived Inbred MSM/Ms Mice"

_biomedicines, 2022, doi:10.3390/biomedicines10092221_

Round 1
Reviewer 1 Report
This paper produced ahl3 congenic mice based on the MSM/Ms model of Japanese wild mice and identified the role of ahl3 and ahl10 loci with potential associations with age-related hearing loss resistance. This article has quite an interesting and novel theme and is well written.
Some suggestions are as follows:
1. “significance was caused” (in many places in the text)? There seems to be no causal relationship. Would it be better to change it to “present”?
2. Line 447: “Figure 7B”? -> Figure 7C. And there is no corresponding Figure 7C on page 14.
3. Line 684-686: There was no comparison of hearing test results for the B6-Chr12CMSM and B6-Chr17MSM strain in Figure 7.
4. Line 740 – 743: Please add references to “genomic compatibility caused by the transfer of MSM-derived genomic regions”.
5. Several spelling and grammar mistakes should be corrected.
Reviewer 2 Report
This is a thorough study of loci contributing to resistance to the age-related hearing loss caused by the Cdh23<ahl> allele in the C57BL/6 strain. There is a lot of work presented here, and identifying loci and candidate genes for resistance to ARHL is very important. However, the manuscript needs some editing for clarity, since some sentences were very hard to parse, and some details are unclear. In addition, as a general note, I think standard deviation would be better than confidence interval for the ABR plot error bars. Standard deviation shows the spread of the actual data collected, whereas confidence interval shows the certainty of the estimate of the mean from the entire population. The rationale for using this is not described, nor are the assumptions required for using a confidence interval discussed or justified.
Corrections/suggestions:
L54,55 - sentence unclear, please rephrase. Is the intention to say "the reduction of genetic and phenotypic heterogeneity is particularly advantageous for a forwards genetics approach to hearing loss"?
L58-60 - why does the forward genetics approach in mice eliminate the effects of dominant risk alleles in hearing loss? Please clarify what is being said here.
pL76 - what is meant by "loss of hair cell delay"?
L96 - it would be helpful here to mention which chromosome Cdh23 is on.
L146 - is there any reason to believe that the hearing of C57BL/6 mice differs between the sexes?
L193-6 - does this mean that three comparisons were carried out, one between each pair of groups? Or is each mouse only allocated to one group? Please clarify.
L275-6 - IHC and OHC counts were each measured at four points along the cochlea, so why are only two averages given here? The differences clearly vary across the length of the cochlea, from Fig 2, so this is confusing. If the average differences along the cochlea have been averaged together again to result in these two figures, I am not sure that is a particularly helpful approach, given the importance of cochlear positioning.
L289-290 - what does "the IHCs of 34.3+-15% and 31.5+-15%" mean? What is "34.3%" and "31.5%" in this context? It can't be the position, since that is referred to in the phrase immediately following.
L298 - Similarly, "OHCs of 88.6+-15%" makes no sense. What are these OHCs? 88.6% of what?
L288-302 - in general, this is a confusing paragraph and hard to read - paragraphs summarising lots of numbers usually are. It's actually much easier to comprehend what's going on from the graphs in Figure 3, so I would recommend rewriting this paragraph to state more simply what is visible, and referring to Figure 3.
L312-313 - Which difference in IHC survival is significant between 2 and 10 months? Please mark on the graph in Fig 4, if there is a significant difference, although I can't see where one could be. It's very odd if there is a significant difference between 2 and 10 months, but not between 2 and 18, or 2 and 24 months.
L334 - Again, "OHCs of 93.6+-3.9%" makes no sense to me.
There is no legend for panels C and D of Figure 4, please add. Also, it would be good to have SEMs from 2 months old to compare the 24 month old hair cell to.
L338-9 - Some of the stereocilia bundles in panel C look distinctly abnormal, so I disagree with this observation. It would be good to make a count of missing, perfectly normal and somewhat misshapen stereocilia bundles across the specified interval (16kHz), and to compare that to what is seen in the 2 month old mouse.
L403-405 - I don't understand this sentence. What is transferred to the MSM-derived region? What is meant by "and strains were detected in the ABR thresholds at 4kHz"? I can't see where the rest of that clause comes from. Please rephrase for clarity.
L407-408 - How can the differences exhibit a similar tendency? A similar tendency to what?
L411-413 - Again, the syntax of this sentence makes it very hard to follow what is being said. I suggest not including the phrases about "transferred to the MSM-derived region" (here or anywhere else, in fact). Firstly, if I understand it correctly, it is the region which is transferred, not transferred to, so I think the grammar needs checking, and secondly (and more importantly), it makes for an unwieldy sentence. It would be better to simply refer to the strains by their names, since they have been more than adequately described in L373-380 and Fig 6A.
L415-416 - What are "suppression effects of increases in the ABR thresholds"? What are the increases in ABR thresholds suppressing?
L437-440 - sentence should be rephrased for clarity; I suggest "In this process, we identified that a B6-Chr12C<MSM> subconsomic strain, in which an approximately 69.8Mb genomic region from the MSM chromosome 12 is transferred to the B6 strain," or similar.
L524 - What is meant by "scarcely" as a description of how the ABR threshold distribution compared to a Gaussian distribution?
L538-539 - I think this should say that the MSM-derived 10.1Mb region was transferred to B6, rather than that the congenic strain was transferred to the MSM-derived 10.1Mb region, which is what it currently says.
L571 - which datasets were examined in the gEAR? Also, several interesting expression patterns are described in the discussion (section 4.4), so it seems wrong to say that there were no highly expressed genes in cochlear cells.
L624 - what is "respectively" doing here? It seems superfluous.
L627-629 - Confusing sentence. "Even though many OHCs are lost from the cochlea, a decrease in DPOAE amplitudes may be avoided if the cochlear OHCs are maintained" - but many OHCs are lost, as has just been stated. Please clarify the intended meaning.
L649 - this sentence says the B6.MSM-ahl3/2 congenic strain was transferred to the MSM-derived 6.3Mb region, which makes no sense, please rephrase.
Round 2
Reviewer 1 Report
I recommend accepting it in its current form.